# Preliminary Research on Planning of Decentralizing Ancient Towns in Small-Scale Famous Historic and Cultural Cities with a Case Study of Tingchow County, Fujian Province

**Min Yin [1], Jiangang Xu [1],\*** and **Zhongyuan Yang [2]**

[1]  School of Architecture and Urban Planning, Nanjing University, Nanjing 210093, China;
   dg1536010@small.nju.edu.cn
[2]  College of Tourism Management, Chaohu University, Hefei 230000, China; faye601@126.com
\*  Correspondence: xjg129@sina.com

**Abstract:** The urban planning industry has always been concerned about conserving and developing historic cities in a sustainable and balanced way. However, unreasonable planning and accumulative effects brought by rapid urbanization prevent the conservation of small-scale famous historic and cultural cities. Taking Tingchow county as an example, this paper focused on sustainable development and the Historic Urban Landscape Approach, and determined the urban functions and specific tasks of various planning of its ancient town with the help of public opinions. This paper mainly aimed at providing guidance on urban decentralization from two perspectives. Firstly, it compared the types of land use and its ratio among famous cities of similar scales, and results showed that it is advisable to reduce three-class residential land use and unnecessary administrative functions. Secondly, it estimated the moderate resident population in different degrees of development, and calculated the upper limit of resource space bearing capacity (REBC) of scenic spots under the guidance of sustainable tourism. Results showed that it is recommended to decentralize and resettle 20%~30% of the resident population, and to control the tourist population below 12,000 per day. As the preliminary work of planning, this paper focused on the scientific planning and availability of decentralization, and reflected an expectation for the mode of public participation and quantitative planning.

**Keywords:** historic cities; urban conservation and development; functional decentralization; population dispersion; sustainable tourism

## 1. Introduction

In May 1964, the Second International Congress of Architects and Technicians of Historic Monuments (ICOM) approved the International Charter for the Conservation and Restoration of Monuments and Sites, that is, namely Venice Charter, which affirmed the value of historic buildings and sites. Fifty years later, the awareness to protect historic cities was gradually shaped in the policy-making circle, and relevant laws were perfected correspondingly [1]. In September 1993, the World Health Organization (WHO) was founded in Fez, Morocco, and the concept of Heritage City was then put forward as an expansion of cultural heritage. Cities, regarded as part of heritage, started to be conserved. Similar to the concept of Heritage City, China owns 134 state-list famous historic and cultural cities (in short, famous cities). The Cultural Relics Protection Law of PRC issued in 1982 clearly stated that cities with an unusual wealth of cultural relics of high historic value and major revolutionary significance shall be recommended to the State Council, and to be approved and announced as famous cities of historic and cultural value [2]. After that, the conservation of historic

and cultural famous cities has played a crucial role in China's urban planning, with an increasing focus on sustainable development.

Due to historic reasons, ancient towns in these famous cities are often the center of the city, with characteristics of significant accumulative effects, excessively accumulated functions, old public services and dense alleys [3,4]. Famous cities are divided into seven types, including ancient capitals, cities with traditional characteristics, cities with famous landscapes, cities with local characteristics, cities with modern historic sites, and cities with special functions. Cities having an edge with their locations often have various features [5]. Thus, these ancient towns not only retain the traditional structure, but also are the places where ancient local governments and numerous historic sites lay. However, since 1800, the urbanization of the entire world has accelerated [6]. The increase of population in these ancient towns, caused traditional buildings inherited from the ancient time to be randomly constructed and segmented, thus bringing about problems like overwhelmed traditional alleys and poor living quality [7,8]. Francesco Bandarin and Ron Van Oers (2013) believed that the population would continue growing and thereby bringing a suite of challenges as cities, in particular historic cities, underwent changes [9] (pp.1–10).

Before the planning on protecting famous cities was determined in the late 1900s, to address the above urban problems, the majority of China's ancient towns received large-scale demolition and reconstruction. During this time, urban development focused on production and construction, and urban renewal went with no guidance [10,11]. Certain cities solely retained important nodes of buildings, while 90% of cities were razed and tower blocks with dense residents were built in ancient towns [12]. This rough renovation ignored the urban landscape, and the natural patterns of old cities were broken due to the increasing height of buildings. Along with the later rapid urban sprawl, some separated residential communities in the periphery of ancient towns were built, but most of them struggled to cater to residents' demands quickly [13]. Ancient areas still attracted migrant people coming here for medicine, education and work. Therefore, roads had to be broadened due to traffic pressure, such as morning and evening rush hours. Residents' self-constructions and self-renovations of traditional buildings destroyed the historic landscape in ancient towns. It can be clearly seen that uncontrolled urban construction and building of dormitory towns are not effective enough to deal with crowded people and protect historic landscapes. In addition, terrible living conditions will prevent residents from protecting historic sites. These problems, if not properly handled, will hamper the conservation of historic landscapes, not to mention the sustainable development of ancient towns.

In 2005, along with the trend of sustainable development, United Nations Educational, Scientific and Cultural Organization (UNESCO) put forward the concept of Historic Urban Landscape (HUL), aimed at better managing historic cities in the context of dynamic updates [14]. It also reminded us to care about the updates of the whole historic cities and lives of residents, rather than just focusing on buildings or monuments isolated from their cultural context and setting [15]. In 2010, based on the summary and expansion of the protection methods in Venice Charter, UNESCO laid out the Recommendation on the Historic Urban Landscape. Apart from giving the specific definition of the HUL approach, it also described, in detail, its methods and tools. Namely, as the policy and strategy of sustainable development, the HUL approach emphasized the significance of society, culture and economy on the protection of urban spirit, the comprehensive sustainable development of historic cities, and multi-side planning and cooperation for conservation work [16,17]. The proposal of this concept indicated that we had started to reflect on static and isolated conservation. The sustainable development of famous cities is no longer the retainment of historic areas. To improve livability and maintain sustainable economic development, we should seize the chances for development and renovate public spaces [9] (pp.1–20). The transformation of this concept demands adjustments of current policies, and planning is also needed for improvement. However, though planners try to encourage the development of famous cities, they have no way to start dealing with complicated functions and over-accumulated populations in ancient towns [18]. Besides, vague planning and barriers between plans impede the conservation of famous cities. In this respect, we attempted to conduct a

preliminary study on planning and discuss effective methods to decentralize functions of ancient towns in famous cities. Decentralization usually includes two aspects: function and population. Function decentralization leads to spatial changes of population distribution, and migration of population brings about recombination of functions.

Most cities will undergo the decentralization of urban functions and population dispersion after experiencing accumulation and high urbanization [19] (p. 8). However, relevant studies on decentralization in China mainly focus on famous megacities like Beijing and Shanghai. However, along with the stimulation of tourism and high-speed railway, small-scale cities (cities with a population of less than 500,000, announced by the State Council [20]) have new chances. New services, such as tourism, have motivated these famous cities to dynamize the lost places and conduct comprehensive conservation [21]. Famous cities at this level account for one third of the whole. Compared with large cities or megacities, they lack experts for management and advanced planning, therefore they are often unable to handle the abruptly accelerated urbanization. Tingchow, a county-level mountainous city in the southwest of Fujian province, as one of the second batch of famous cities (approved and announced in 1994), is facing this dilemma [22]. Driven by modernization, problems like the rapid growth of population in ancient towns, limited land for construction, and environmental problems have emerged there.

As sustainable development and decentralization of ancient towns' functions are essential, this paper listed and summarized the relevant planning of decentralizing historic cities' functions in China and western countries. Then, based on the case study of Tingchow, we conducted preliminary study on planning of decentralizing ancient towns' functions. After summarizing the results of field research and public participation in Tingchow, we determined the functional orientation of Tingchow ancient town and planned functional decentralization in the context of dynamic updates. The indices of land use, generated from the study, were then applied to calculate and control the volume of inhabitants and migrant population in ancient towns, which can serve population dispersion. The conclusion of the study and prospect for future development are presented at the end, which can provide guidance for the following work on planning.

## 2. Literature Review of Relevant Theory and Planning Experiences of Decentralizing Urban Ancient Towns

Decentralizing ancient areas is not a new topic. In the early 1900s, with many people flowing to cities, a series of urban problems emerged in countries all over the world [18] (p. 9). It was the time when cities started to enforce the practice of population and industry dispersion, and relevant studies carried out by pioneer architects and social activists were developed accordingly. One of the milestones was Garden Cities of Tomorrow, published by Ebenezer Howard in 1898 [23]. The book envisages a sustainable future city with limited population, strict functional layout and abundant green space. This research has a far-reaching impact on the theory and practice of planning for later generations. In the 1940s, E Saarinen, a Finnish architect, proposed the Theory of Organic Decentralization [24]. After World War II, many megacities used these theories as the guidance for planning, for the purpose of adjusting the spatial layout and development strategies of cities.

According to these theories, historic cities such as London, Paris and Tokyo have established many satellite cities to disperse the population and maintain the ancient cities' landscape. The Greater London Plan was completed in 1945 [25]. The Greater London Plan reorganized a spatial structure of the metropolitan region, taking London as its center, based on organic decentralization, and mapped out a dozen new "half-dependent" counties, thereby decentralizing residential and employment functions in the central area of London. These counties contained one-third of the population in the central area, and shared the pressure of post-war urban reconstructions. The Greater Paris plan, completed in 1965, decentralized urban functions at a larger scale, and changed the accumulative trend of people and jobs toward its central area. Along the downstream of the Seine, a metropolitan area formed, with a series of sub-centers. In addition, the original monocentric urban layout changed into a multi-centric one [26].

The planning of decentralization of Asian cities started relatively late. Under the huge pressure on transportation, environment and housing brought by the boom of population and economy in the ancient area, Tokyo, during 1955–1995, put forward several plans, such as National Capital Region Planning (1956), The Second General Plan on Capital Region (1968), Planning on Capital Reform, and the Fourth General Plan on Capital Region, etc., and some compulsory national regulations, such as Control on Industrial and Other Facilitates in Build-up Capital Region, and Rules of Renovation to Development Zones in Capital Region, etc., was enacted to enhance the top-level planning at the national level, and promote gradual decentralization of various facilities once accumulated in central Tokyo, including industry, education, administration, wholesaling, warehousing, and circulation, thereby decentralizing people in central area [27].

While in China, its work in urban decentralization at first mainly focused on ancient towns (that is, central areas) in megacities like Beijing. During the early years of China, the government convened experts from several fields such as western-educated architects and engineers, to participate in the urban planning, in which the Theory of Organic Decentralization was emphasized [28]. The "Mother-son towns (similar to satellite towns)" and "distributed and clustered" layouts were determined by Beijing Master Planning (1958), and it set the tone for the framework of development in the next 50 years [29]. Since then, the decentralization of the city center and the conservation of the Forbidden City (ancient area) is one of the core missions of the urban planning of Beijing. In recent years, Municipal Master Plan of Beijing (2016–2035), put forward that the special operation of "upgrading through function transfer and remediation" should adhere to Beijing strategic planning [30,31]. At the same time, it can be seen from the planning that, according to the theory of urban sustainability, the urban functional orientation keeps changing all the time; and therefore, we should respect the ecology and conduct comprehensive conservation when considering urban development and functional decentralization. After that, the Beijing government has issued various regulations to control population, such as promoting the phase-out of urban functions, enhancing the development of urban-rural fringe area, renovating shantytown renovation, and transferring certain industrial and preponderant sources, etc. Therefore, it changed and improved the industrial structure and layout of its center, and mostly focused on decentralization of population and functions.

Urban decentralization has hardly been applied to small-scale cities, as well as ancient towns, and is in need of more attention and comprehensive studies. However, Germany has been focusing on the management of small-scale cities, which gives reference to this study. There are only four cities in Germany whose population are over a million (and the one with the most population has less than four million people). The majority live in small-scale cities with 100,000–500,000 people. Cities with 100,000–200,000 people in Germany can be regarded as an "integration of urban and rural ecology" [32]. The success of Germany lies in reasonable planning, facility allocation and taxation distribution. It obeys market rules and relies on the construction of railways to provide certain benefits for small-scale cities.

## 3. Methodology

This paper focused on small-scale famous cities, as they have relatively small populations but outdated planning theory and economy. Considering these drawbacks, the study selected a typical small-scale city as the case and combined qualitative and quantitative methods to plan for the decentralization of ancient towns in these cities.

### 3.1. Background of the Case Study

Tingchow, a mountainous city in Fujian province, was determined as a county in Han Dynasty. From Tang Dynasty to Qing Dynasty, Tingchow was the political and cultural center of southeast China, as well as an important populated place for the Hakkas, thereby credited as "the capital of the Hakkas." In modern China, Tingchow was an important revolutionary base area. As the economic center of Jiangxi–Fujian Soviet, it was called "Little Shanghai" [33]. Due to its integrated human and

natural landscape, Tingchow, together with the Ancient Town of Fenghuang in Hunan province, was considered as one of the most beautiful mountainous cities in China by Rewi Alley. Ranking among the second group of famous cities, Tingchow owns diversified cultures, unique local characteristics and masses of superior resources for tourism.

Tingchow has a population of less than 400,000 (Table 1), among which nearly 80,000 are urban population. According to the sixth census, people aged 65 and above accounted for 9.57% of the population, higher than the aging standard. As is shown in a survey conducted by Nanjing University in 2016, among the 3804 families, those with an annual income of over 100,000 accounted for less than 10% of the whole. A field research conducted from 2013 to 2017 showed that shabby living conditions forced young people to leave ancient towns; traditional houses became tenements, and even eight solitary elderly people lived together in one house; ancient buildings, mainly dwelling houses, were burnt, torn down, or self-renovated; facilities congested in the sole trunk road, causing huge commuting pressure (Figure 1). The loss of young people challenged the succession of culture, and over-congestion led to a deteriorating environment, but the government was unable to improve facilities and make timely remedy for destroyed heritage due to the depressed economy.

**Table 1.** Population in Tingchow.

|  | **Resident Population** | **Ratio of Migrant Population** |
| --- | --- | --- |
| Sum of Urban Population | 79,802 | 40.76% |
| Sum of County's Population | 393,390 | 24.10% |

Source: the 6th National Census (2010) by National Bureau of Statistics of China.

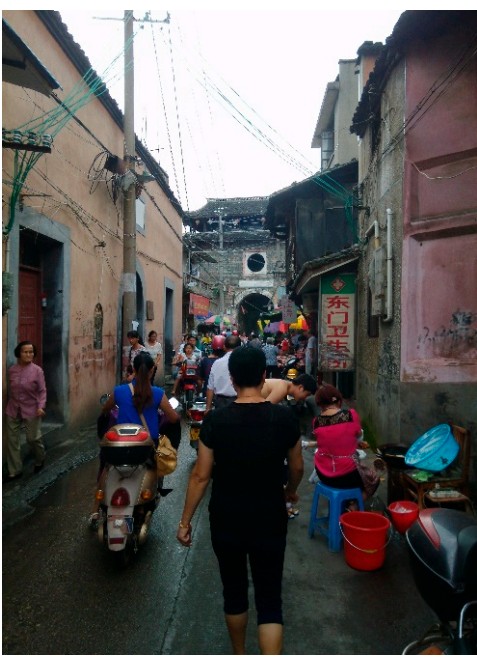

**Figure 1.** Tingchow Ancient Town (by Author).

Meanwhile, Tingchow started late in tourism. Its tourism revenue and tourist population are still at a low level compared with other areas in Fujian province. However, benefiting from the increasingly convenient transportation and national policies, Tingchow earned a tourism revenue in 2013 that was nearly 10 times higher than that in 2008 (Figure 2), receiving more than 1.2 million tourists per year. The Tingchow High-speed Railway Station on Ganzhou-Longyan Railway opened in 2015, and thereby potential tourists doubled in its 3-hour coverage (Figure 3). The increase of tourist population will

occupy more spatial resources. Therefore, we should seize the chance for tourism development while be cautious about the explosive population growth brought by the over-expanded market.

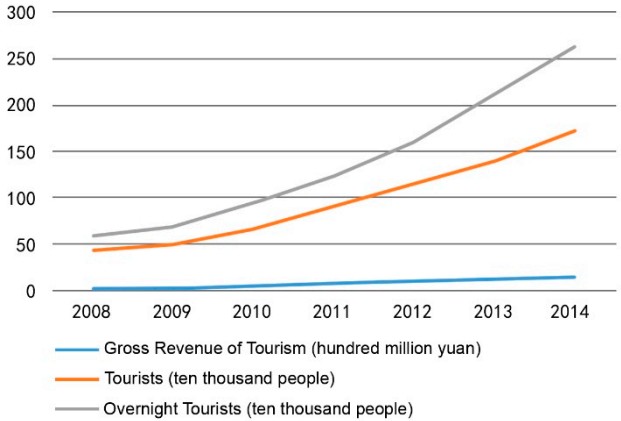

**Figure 2.** Tourism development in Tingchow County from 2008 to 2014 (Source from: Tingchow Statistical Bureau).

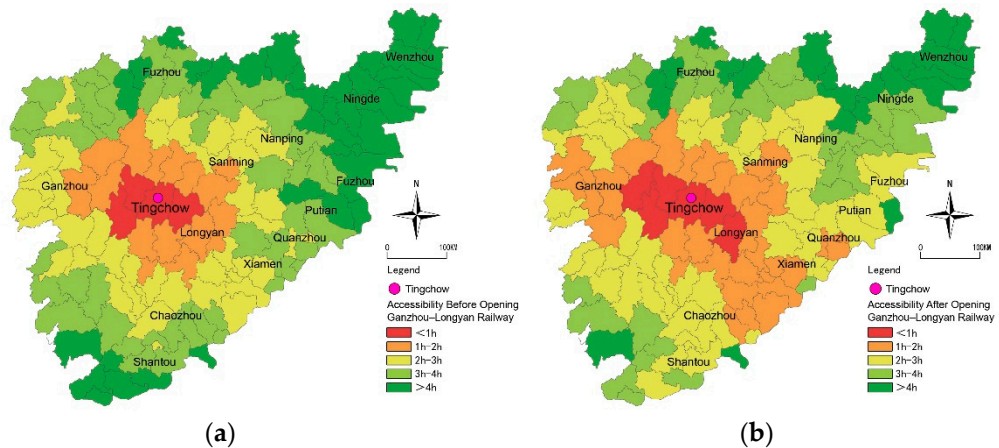

(**a**)　　　　　　　　　　　　　　　　　(**b**)

**Figure 3. (a)** Analysis of Tingchow's accessibility before opening Tingchow high-speed railway station on Ganzhou-Longyan Railway; **(b)** analysis of Tingchow's accessibility after opening Tingchow high-speed railway station on Ganzhou-Longyan Railway (by Author).

### 3.2. Research Framework and Methods

As is mentioned above, ancient towns integrate functions of administration, culture, sanitation and education because of historic reasons. Complicated identities cause overloading. Therefore, if we want to decentralize the ancient towns and improve services, the theory of top-level development and position of famous cities should be determined mainly by qualitative study, ahead of all the planning (Figure 4). With the HUL approach as reference, multi-party participations, especially opinions of local residents and non-government organizations, are recommended to be considered when determining developing strategies [19] (pp. 11). A large amount of literature and social survey data are needed to analyze ancient towns' complicated identities. Large amounts of data were collected in three social surveys which School of Architecture and Urban Planning of Nanjing University organized in 2013–2016, and this can be taken as a reference for preliminary study on planning. At the same time, literature review, applied in the preliminary research on planning, clarifies functions of various statutory and non-statutory planning, such as planning for famous cities' conservation, urban master plans and planning for developing tourism during the top-level strategy making.

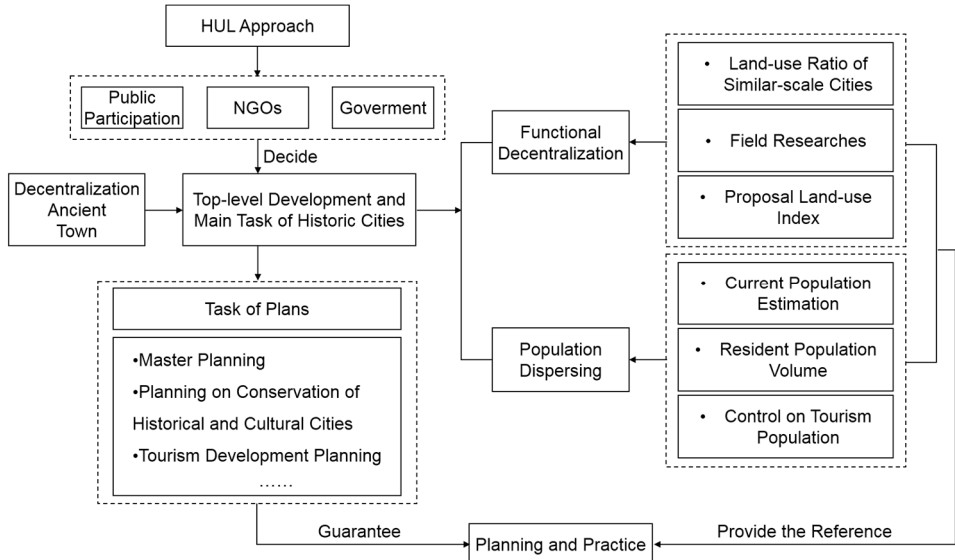

**Figure 4.** Research framework (by Author).

Second, the goal and supportive strategies of functional decentralization should be raised in planning. Through the comparison and analysis on several small-scale but developed historic and cultural famous cities, we developed a proper ratio of land use, and provided a specific functional distribution of lands and quantity indices. At last, we gave advice for planning decentralization at all levels and provided reference for land use planning.

Third, we should set a goal of population dispersion, including the dual control on the volume of resident and tourist populations. To ensure precise decentralization, quantitative methods are mainly applied to generate specific and thereby manageable data for conclusion. Early studies on urban population volume were primarily aimed at expansion and development, pursuing the maximum population volume. Recently, the Theory of Optimum Population was developed, and thereby the studies on population volume tend to generate a more proper index. The population that needs to be dispersed in planning should equal current population minus optimal population. Therefore, current population is needed to be dispersed precisely. Population data in China mainly come from the census, but as ancient towns do not belong to administrative areas, an accurate population number is not available there and such data mask underlying individual population distributions [34]. We adopted and optimized the model of population density built by Xu Jianggang et al. (2015) to estimate current populations in ancient towns, and calculated optimal residential population volume via residential lands, thereby generalizing the optimal range for population dispersion [35] (p. 84). In addition, to optimize population volume, we should take strict control on actions such as establishing new buildings and reconstructing old buildings. It is also advisable to renovate residential lands and reduce floor area ratio, in order to reduce population volume. The decentralization of resident populations does not mean to solely develop services. Tourism should not be developed at the cost of compressing the living space of indigenous people [36] (pp.45–47). Thus, we should combine the research on population dispersion with that on migrant population volume, based on the upper limit of Resource Space Bearing Capacity, and control the number of tourist population number and scenic spots [37]. The specific models are as follows

- Current Population Estimation Model

In the population-estimating model, as residents people generally have their own house, the distribution of population is consistent with the area of buildings [35] (pp. 85–87). Based on topographic maps, remote sensing images and field studies, residential lands are divided into A-class, B-class and C-class according to living conditions (the A-class is the residential land with relatively good quality,

the B-class is that with ordinary quality, and C-class represents the bad environmental conditions). The total area of buildings in different lands is also calculated. If we assume the per capita floor area of all classes are similar, data, such as floor area and its ratio of different classes of land, can be used to estimate population density. The model supposed that $n$ types of residential land existed in research area, $B_i$ ($i = 1,2, \ldots , n$) was the reciprocal value of per capita floor area. Research area was divided into $m$ administrative units, and statistical population of each unit was $P_j$ ($j = 1,2, \ldots ,m$). The total floor area was $A_{ij}$ ($i = 1,2, \ldots , n; j = 1,2, \ldots , m$). Then it followed:

$$A_{11}B_1 + A_{12}B_2 + \cdots + A_{1n}B_n = P_1$$

$$A_{21}B_1 + A_{22}B_2 + \cdots + A_{2n}B_n = P_2 \vdots A_{m1}B_1 + A_{m2}B_2 + \cdots + A_{mn}B_n = P_m \tag{1}$$

The reciprocal value of per capita floor area $B_i$ (i $= 1,2, \ldots , n$) can be generated from the above equations. After allotting errors in proportion to floor area to all classes of residential lands, corrected $B'_{ij}$ could be obtained. Based on $S_f$, area of residential land on block F and $P_f$, population of block F, we can get $D_f$, the population density in different kinds of residential lands on block F:

$$D_f = \frac{P_f}{S_f} = \frac{\sum B'_{ij} A_{ij-f}}{S_f} \tag{2}$$

- Resident Population Model

To conduct population dispersion, the resident population model was applied to estimate population volumes in ancient towns. According to Code for Classification of Urban Land Use and Planning Standards of Development Land and specific situations in ancient towns, we divided the residential lands into $n$ areas and established flexible planning. $R_i$ ($i = 1,2, \ldots , n$) represented the moderate volume ratios of various residential lands, $M_i$ was the area of residential land, and $C_i$ represented the per capita floor area. Therefore, the final population volume V can be calculated:

$$V = \frac{M_1 R_1}{C_1} + \frac{M_2 R_2}{C_2} + \cdots \frac{M_n R_n}{C_n} = \sum_n^{i=1} \frac{M_i R_i}{C_i}. \tag{3}$$

Population of dispersion was $V''$, and

$$V'' = V_{\text{current}} - V. \tag{4}$$

The extent of population dispersion generated in different ancient towns varied as they were at different developing stages.

- Resource Space Bearing Capacity Model

REBC represents the number of tourist volume influenced by viewing time of tourism resources and spatial requirements during a certain amount of time (in an instant or a day) [38–41]. It is calculated mainly by gross volume model and flux-flow velocity model. As Tingchow's tourism is in the early stage, we adopted the gross volume model here, that is:

$$D_m = S/d, \tag{5}$$

$$D_a = D_m(T/t). \tag{6}$$

It can be seen from Equations (5) and (6): $D_m$ represented the tourist volume in an instant in sites (unit: people), $D_a$ represented the sum of tourist volume in a day. $S$ represented area for sightseeing (unit: m$^2$). $d$ represented the best per capita spatial area for sightseeing (unit: m$^2$/person). $t$ represented

the average sightseeing time, which varied as the scale and level of sites differ. *T* represented the effective time for sightseeing, that is, the opening hours of sites.

## 4. Results and Discussion

### 4.1. Sustainable Tourism: New Orientation of Tingchow's Urban Functions

Hu Hao, a geographer in Chinese Academy of Sciences, pointed out that famous cities are the centers of cultural development at a certain degree, and they have obligations and responsibilities to provide cultural products and services for cultural tourism [42]. UNESCO believed that the cultural diversity of famous cities is important for the development of humanity, society and economy, and the conservation of diversity and development of tourism will be mutually beneficial [43]. Cultural diversity is the precondition of superior resources for tourism, and developing tourism can activate culture and services, and accelerate the reform of outdated industries. However, without macro-policy control, the expansion of market will cause an accumulation of similar spatial functions, as well as the unreasonable volume and structure of populations. In 1993, World Heritage Center (WHC) raised the concept of sustainable tourism [44]. To activate tangible and intangible local culture, it encouraged the coordination of preserving heritages, developing tourism and prospering communities, based on protecting local core values and respecting residents' rights. In 2011, the HUL proposal demonstrated that many economic developments help alleviate urban poverty and promote social development [45]. New functions, driven by sustainable tourism, play an important role in promoting economy and increasing social welfare. It promotes the conservation of cultural heritage, ensures economic development of cities and retains the social diversity as well as residential functions. If we cannot seize the chances, cities will lose their sustainability and livability. Due to what improper urban developments do to heritage areas, an unredeemable loss will be caused to future generations.

Based on the above advanced theories, Nanjing University conducted a survey to investigate public awareness on famous city conservation in July 2015. Among 433 validated questionnaires received (Table 2), 410 people were in favor of developing tourism (including 260 people strongly agreeing to developing tourism and 150 generally agreeing).

**Table 2.** Public awareness on famous city conservation.

|  | Strongly Agree | Agree | Generally Agree | Disagree | Strongly Disagree |
|---|---|---|---|---|---|
| In favor of tourism development and tourists | 260 | 150 | 22 | 0 | 1 |
| In favor of policies for tourism development | 229 | 163 | 36 | 4 | 1 |
| Willing to work in tourism industry | 141 | 171 | 102 | 18 | 1 |
| Tourism can improve economy | 215 | 191 | 26 | 0 | 1 |
| Tourism can bring more jobs | 190 | 190 | 48 | 2 | 2 |

Source: social survey's finding (by our group).

Shortly afterwards, in November 2016, more than 30 deputies of non-government organizations (NGO) from 26 famous cities, including Zhengding county of Hebei province, She county of Anhui province, and Xitang town of Zhejiang province, together with over 10 famous experts of bed-and-breakfast from Mogan Mountain of Zhejiang province, Taiwan Rural Accommodation Association, as well as other areas, attended the talk of Conservation of Famous Cities and Tourism Development held by Tingchow county. They jointly made Tingchow Declaration for Conservation of Famous Cities. The declaration mentioned that the ancient town in Tingchow was its greatest resource for tourism development, and we should focus on the point to conserve its cultural heritage in multiple ways and inspire its vitality. In December 2016, Tingchow signed *Framework Agreement for Strategic Cooperation* with Nanjing University. The government, together with experts in architecture colleges,

set the framework of top-level development and the main task Tingchow should take—developing tourism in a sustainable way. The framework emphasized that multiple programs should be carried out at the same time to clarify specific functions of ancient towns, and control proper ratio of different land use based on requirements in planning.

According to the complex requirement summarized by WHO—heritage conservation, tourism development and prosperous communities, it is suggested to integrate various regulations and cooperation of various departments, to break the barriers between different fields and departments and transform public awareness. This study attempted to take control of the layout and scale of land use in core areas of ancient towns to ensure sustainable development, which is the common task of planning. Additionally, the plans were required to obey top-level framework and optimize urban structure comprehensively (Table 3).

**Table 3.** The comparison of specific content of plans.

| Name of Plans | Task of Plans | Proposed Planning Content | Legally Protected or Not |
|---|---|---|---|
| Master Plan | Set goals and strategies for urban development according to local characteristics, and promote comprehensive, coordinated and sustainable development. | Make new orientation for Tingchow—the state-list famous city, the capital of the Hakkas, and the modern tourist city with ecological, industrial and trade characteristics. | Statutory Planning |
| Planning on Conservation of Historic and Cultural Famous Cities | Coordinate conservation with development to set principles, content and focus of conservation, define the range of conservation, and put forward strategies of protection. | Regard the tangible space and intangible culture of ancient towns as the main targets of conservation, decentralized non-famous-city functions according to the geographic structure of "historic buildings—historic areas—harmonic areas." | Statutory Planning |
| Tourism Development Planning | Set targets of tourism based on market changes, and arrange elements for tourism development to realize targets under special conditions. | Suggest that modern services, led by cultural tourism, should be built according to the overall plan raised by the Central Committee to develop economy, politics, culture, social progress and ecology | Non-Statutory Planning |

Source: review and summary by Author.

### 4.2. Adjustments in Gross Size of Land Use Guided by the Theory of Functional Decentralization

Specifically speaking, the preliminary study on these three plans will compare several developed famous cities at similar scales, including Wuzhen, Fenghuang County, Old Town of Lijiang, Zhengding, and analyze the types and ratios of land use of their ancient town. It is concluded that the proper ratio of residential land should be around 40%, business land around 30%, green field more than 10%, and administrative land relatively less (Figure 5). It can be seen from the comparison that Tingchow's ancient town (within wall relics and Shuidong district) covers an area of 2.32 m$^2$, with too little business land, too much residential and administrative lands, and an especially high ratio of C-class residential land. After the household study and interviews with relevant departments, we provided following advice on accessible land use planning (Figure 6).

As is shown in the results, the three plans should state that the lay-out of ancient towns should not emphasize too much on urban functions, but should focus on residence, tourism and commerce. According to our team's household-by-household survey, we selected decayed parts of ancient areas and the buildings or land that should be replaced, and preliminary guidance was provided (Figure 6). Generally speaking, the specific work for decentralization is advised as follows: certain untransformed industrial and storage lands should be reduced to focus on developing modern services; planning of conservation should emphasize that ancient towns, in principle, should decentralize their administrative function (Figure 7) to reduce attractions to the non-tourist floating population instead

of developing infrastructure to attract local people; extra residential land should be no longer available and low-quality residential land needed renovation in order to control population volume (Figure 8). Some types of lands could be retained, but certain functions, such as, senior high department of school and in-patient department of hospital should be transferred to the southern high-speed-railway new ecological city due to their large radiation range. Considering Tingchow's identity of ecological demonstration county, the ratio of greenfield land should be increased to more than 30% to benefit local residents and compensate the lack of sports land in Tingchow (Figure 9). Business land is allowed to increase to 11% to serve tourism (Figure 10). Meanwhile, the Master Plan should provide guidance to properly decentralize functions of ancient towns, as it plans to gradually perfect supporting facilitates in areas surrounding the high-speed railway station. After years of construction, a transport network between the high-speed railway station and the ancient town will be established. To balance residence and work, the government should put forward a series of supporting constructions, preferential policies and compensations for land use, in order to transfer more people and functions to the high-speed railway station area. In addition, the market should provide more benefits to promote the decentralization of industries, enterprises and institutions. It can be seen from the past that satellite towns, if ignoring inner momentum and system for development, cannot provide support for decentralization. A combination of traffic networks, the government and the market is needed to ensure the decentralization of ancient towns.

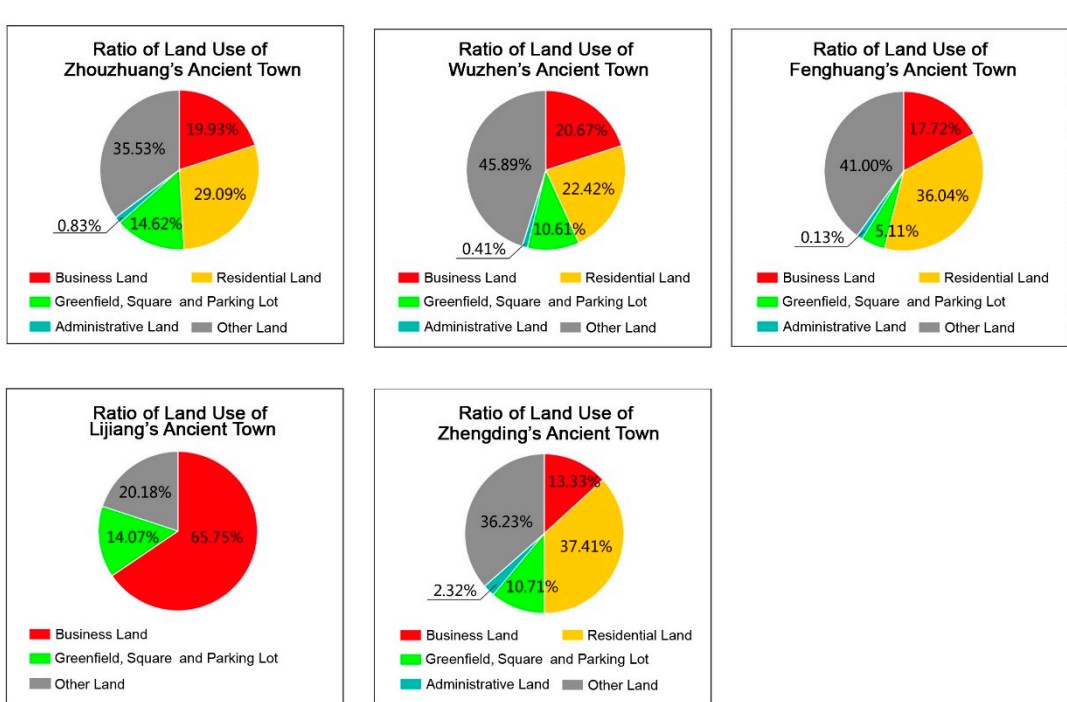

**Figure 5.** Ratio of all types of land use in ancient towns (by our group).

### 4.3. Population Volume Control of Ancient Towns

To realize sustainable tourism, it is fundamental to protect the local ecological environment, and aim at improving urban services. However, a relatively high density of population and aging problem hamper the improvement of cultural environment and living conditions in ancient towns. The conservation and renovation of Tingchow conflict, but cannot live without each other. Therefore, the control on population volume should also consider these two aspects. Based on the research of Sections 4.1 and 4.2, the range of moderate population volume in ancient towns is generated to provide guidance for the planning and other work of population dispersion in ancient towns.

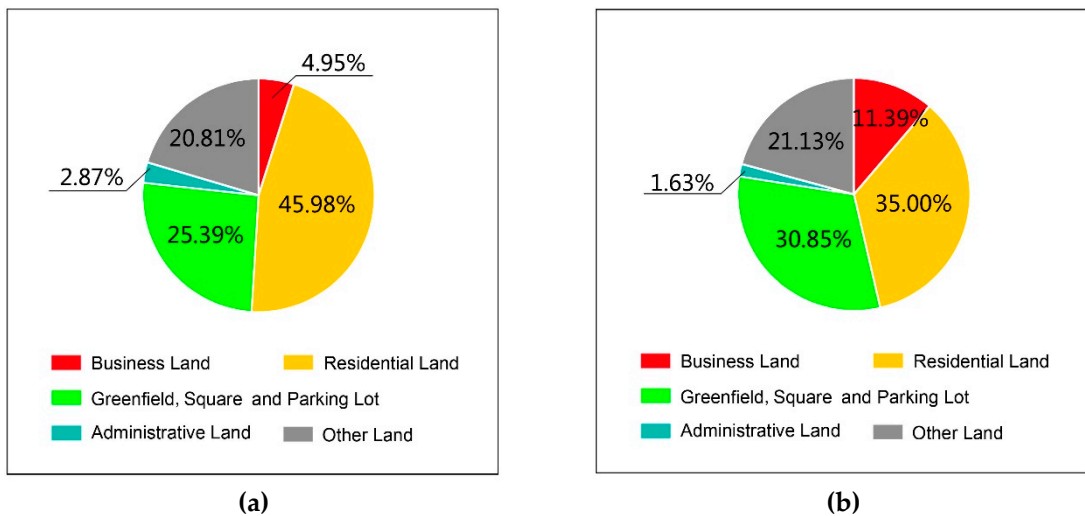

**Figure 6.** (**a**) Current ratio of all types of land use in the ancient town of Tingchow; (**b**) proposed distribution of green land in the ancient town of Tingchow (by Author).

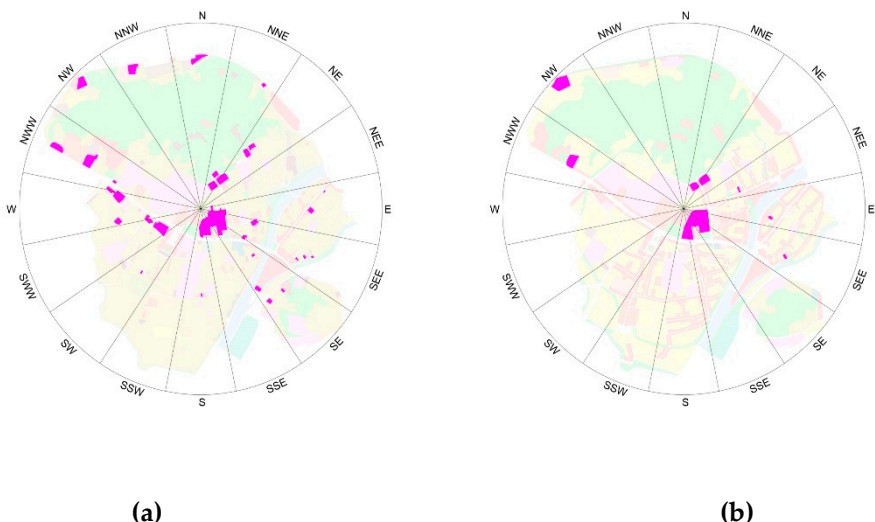

**Figure 7.** (**a**) Current distribution of administrative land in the ancient town of Tingchow; (**b**) proposed distribution of administrative land in the ancient town of Tingchow (by Author).

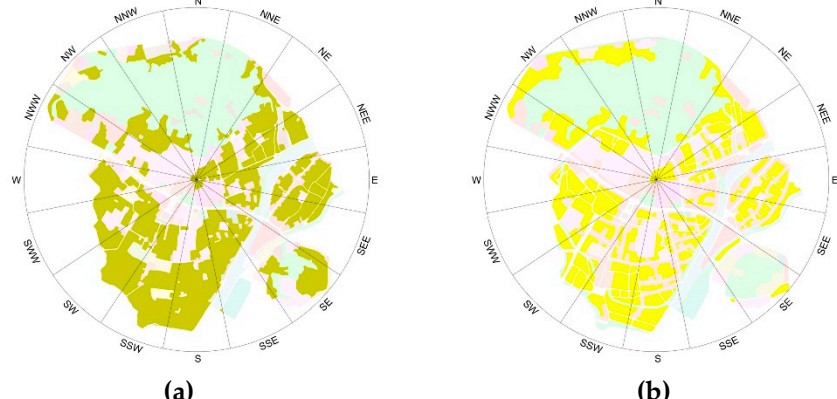

**Figure 8.** (**a**) Current distribution of residential land in the ancient town of Tingchow; (**b**) proposed distribution of residential land in the ancient town of Tingchow (by Author).

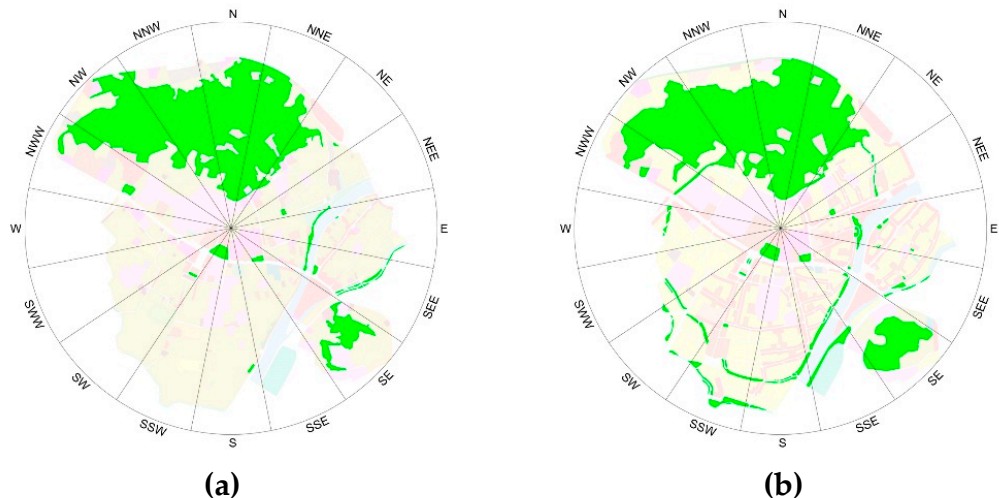

**Figure 9.** (**a**) Current distribution of greenfield land in the ancient town of Tingchow; (**b**) proposed distribution of greenfield land in the ancient town of Tingchow (by Author).

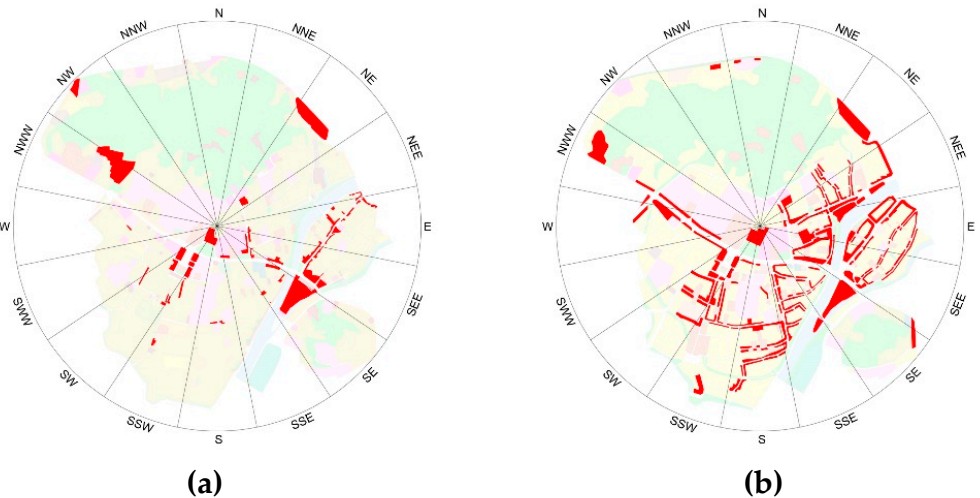

**Figure 10.** (**a**) Current distribution of business land in the ancient town of business; (**b**) proposed distribution of residential land in the ancient town of Tingchow (by Author).

### 4.3.1. Estimation of Present Population

To analyze population distribution in Tingchow's ancient town, we collected census data of Chengguan county, Datong county, taking neighborhood or village committee as an administrative unit (Table 4). Based on topographic maps, remote sensing images and field studies, residential lands are divided into A-class, B-class, C-class and residential lands for farmers.

It can be seen from Table 4 that five village committees including Hongwei village only have residential land for farmers. Therefore, after regression analysis on four forms of residential lands of the above five village committees, it can be calculated that $P = 184.775\ S_4$ ($R^2 = 0.9147$, $p < 0.05$), and $B_4 = 184.775$.

Regression analysis was then conducted to acquire index B. In addition, per capita living space in Tingchow's ancient town was $1/B_1 = 1/B_2 = 62.98$, $1/B_3 = 35.11$. Grids of residential land in different images were extracted to calculate area of buildings. It was calculated that the ancient town had a resident population of 37,247, and certain areas have a high density of 60,000 (Figure 11).

**Table 4.** Population and floor area of various residential lands of neighborhood or village committees in Tingchow county.

| Neighborhood or Village Committees | Resident Population (People) | Floor Area (hm²) | | | |
|---|---|---|---|---|---|
| | | A-class | B-class | C-class | Residential Land for Farmers |
| Committee of Ximen Street | 22,399 | 0.000 | 57.087 | 30.84 | 2.055 |
| Committee of Nanmen Street | 10,185 | 0.000 | 7.867 | 34.595 | 2.573 |
| Committee of Zhongxinba Street | 10,029 | 0.000 | 35.911 | 8.095 | 1.067 |
| Committee of Shuidong Street | 8576 | 0.000 | 7.206 | 14.211 | 0.618 |
| Committee of Yingbei Street | 15,633 | 1.139 | 45.091 | 34.856 | 2.588 |
| Committee of Donemen Street | 12,980 | 0.000 | 64.104 | 19.945 | 2.854 |
| Xingmin Village | 6195 | 0.000 | 16.026 | 0.077 | 8.169 |
| Dongguan Village | 4987 | 0.000 | 11.092 | 3.401 | 0.553 |
| Yinhuang Village | 5503 | 0.090 | 4.949 | 0.235 | 5.156 |
| Huangwu Village | 5536 | 0.000 | 9.175 | 0.996 | 10.338 |
| Jisheng Village | 1379 | 0.000 | 0.154 | 0.000 | 11.065 |
| Hongwei Village | 5853 | 0.000 | 0.000 | 0.000 | 26.183 |
| Caoping Village | 2062 | 0.000 | 0.000 | 0.000 | 3.737 |
| Hongxing Village | 691 | 0.000 | 0.000 | 0.000 | 1.755 |
| Luofang Village | 6676 | 0.000 | 12.686 | 0.000 | 29.180 |
| Liling Village | 1802 | 0.000 | 0.000 | 0.000 | 14.214 |
| Dognjie Village | 3165 | 0.000 | 0.000 | 0.000 | 22.037 |

Source: Tingchow police department.

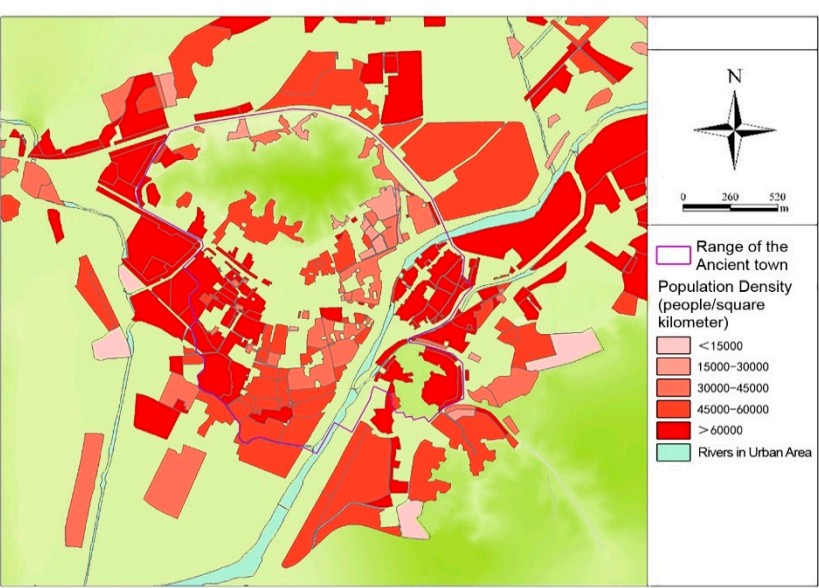

**Figure 11.** Distribution of population density in the ancient town (by Author).

### 4.3.2. Resident Population Volume Optimization

Considering special buildings in famous cities, it is impossible to simply set one unified standard of building height and floor area ratio. To conduct population dispersion and alleviate overcrowded living conditions, we should follow the geographic structure of "historic buildings—historic areas—harmonic areas" and apply different indices to different lands. According to Section 4.2., by 2030, residential land in Tingchow's ancient towns will mainly consist of the B-class, with a few A-class or C-classes. Three schemes to estimate population volume in Tingchow's ancient towns were established according to the different stages of development (Table 5):

**Table 5.** The estimate of population volume of Tingchow's ancient town.

| Scheme | Type of Land Use | Area (hm²) | Floor Area Ratio | Floor Area Per Capita (m²) | Population Volume (Thousand People) | Total Volume (Thousand People) |
|---|---|---|---|---|---|---|
| Top | A-class Residential Land | 0.55 | 1 | 50 | 0.01 | 3.00 |
| | B-class Residential Land in Historic Districts | 55.03 | 1.3 | 40 | 1.79 | |
| | B-class Residential Land in Ordinary Historic Areas | 4.38 | 1.5 | 35 | 0.19 | |
| | B-class Residential Land in Harmonic Areas | 17.39 | 1.6 | 35 | 0.79 | |
| | C-class Residential Land in Harmonic Areas | 4.04 | 1.6 | 30 | 0.22 | |
| Middle | A-class Residential Land | 0.55 | 0.8 | 50 | 0.01 | 2.80 |
| | B-class Residential Land in Historic Districts | 55.03 | 1.2 | 40 | 1.65 | |
| | B-class Residential Land in Ordinary Historic Areas | 4.38 | 1.4 | 35 | 0.18 | |
| | B-class Residential Land in Harmonic Areas | 17.39 | 1.5 | 35 | 0.75 | |
| | C-class Residential Land in Harmonic Areas | 4.04 | 1.5 | 30 | 0.21 | |
| Bottom | A-class Residential Land | 0.55 | 0.8 | 50 | 0.01 | 2.56 |
| | B-class Residential Land in Historic Districts | 55.03 | 1.1 | 40 | 1.51 | |
| | B-class Residential Land in Ordinary Historic Areas | 4.38 | 1.2 | 35 | 0.15 | |
| | B-class Residential Land in Harmonic Areas | 17.39 | 1.4 | 35 | 0.7 | |
| | C-class Residential Land in Harmonic Areas | 4.04 | 1.4 | 30 | 0.19 | |

Source: simulation results by Author.

Top scheme: B-class residential land in historic districts should be less than five stories. During decentralization, renovation should be conducted by a middle-or low-density developing index, ensuring more than 40 m² of floor area per capita. The average floor area ratio of residential area in ordinary historic areas and harmonic areas should be controlled between 1.5 and 1.6, floor area of house per capita should be controlled between 30–35 m², and population volume in ancient towns should be estimated to be nearly 30,000 people. The realization of population dispersion mainly depends on reducing the area of C-class residential land.

Middle scheme: residential land in historic districts should be developed at a low rate of capacity. The average floor area ratio should be set at 1.2, the floor area of house per capita should be 40 m², and floor area of other lands should be controlled between 1.4 and 1.6. In addition, population volume in ancient towns should be nearly 28,000 people. The realization of population dispersion mainly depends on reducing the area of C-class residential land.

Bottom scheme: B-class residential land in historic districts should be controlled according to the environmental standard of A-class constructive land. The average floor area ratio should be set at 1.10, and the floor area of house per capita should be less than 40 m². Floor area of other lands should be controlled between 1.2 and 1.4. In addition, population volume in ancient towns is estimated to be nearly 25,600 people. The realization of population dispersion mainly depends on reducing area of C-class residential land and increasing the floor area of house per capita.

In sum, population volume of ancient towns set in three schemes is estimated to be between 25,600 and 30,000 people. By 2030, various methods like industrial layout and construction of high-speed railway new towns will decentralize 7200 to 11,600 people.

4.3.3. Control on Tourist Population

In Tingchow county, nearly 1/3 of tourism resources are distributed in the ancient town, and 76 historic sites under national protection at or above the county level, with an area of 45,900 m$^2$, are open to the public. If population grows abruptly, it will put pressure on the ancient town's environment. Therefore, besides encouragement on cultural tourism, controlling the scale of tourism is also crucial for preliminary study. Primarily based on the upper limit of resource space bearing capacity (REBC), the study provided guidance to controlling the scale of tourism. It is calculated mainly by gross volume model and flux-flow velocity model. As Tingchow's tourism is in the early stage, we adopted gross volume model here, that is:

It can be seen from Equations (5) and (6): $D_m$ represented the tourist volume in an instant in sites (unit: people), $D_a$ represented the sum of tourist volume in a day. S represented area for sightseeing (unit: m$^2$), which was calculated with 60%–80% of its area in ordinary models. Due to the specialty of famous cities and to protect historic sites, the floating population should be strictly controlled. Therefore, the calculation took 50% of the area of historic sites under national protection, 55% of the area of historic sites under provincial protection, and 60% of the area of historic sites under county's protection. d represented the best per capita spatial area for sightseeing (unit: m$^2$/person). According to the standards of classic gardens (20 m$^2$/ person) and parks (10 m$^2$/ person), d ranged from 10 m$^2$/person to 15 m$^2$/person. t represented the average sightseeing time, which varied as the scale and level of sites differ. T represented the effective time for sightseeing, that is, the opening hours of sites. It can be seen from the research that the average opening hours of sites in Tingchow is 10 h. It can be estimated that the moderate value of daily REBC of sites in the ancient town ranged from 80,229 to 120,344 people (Figure 12).

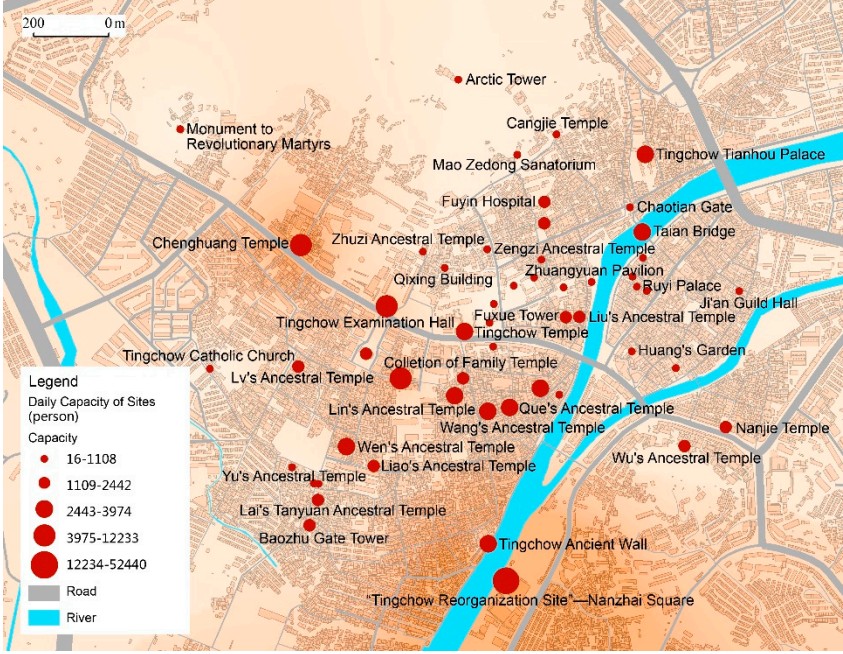

**Figure 12.** Daily resource space bearing capacity (REBC) of sites in Tingchow's ancient town (by Author).

Plans of tourism and the relevant policies should respond to these results. A joint control of many departments is necessary. Based on the above indicators, local governments should take actions including establishing strict regulations for historic sites at national, provincial and county levels, coordinating opening hours of departments and sites and alternately opening sites or changing tour routes. A comprehensive plan to manage new supporting services is needed. In addition, it is necessary to keep the ratio of business in a moderate range according to Section 4.2.

## 5. Conclusions and Vision

Over the last 50 years, we have witnessed the progress on urban conservation and regeneration. However, it is a challenge to protect historic cities in a more sustainable and balanced way rather than freezing them. China owns 134 state-list Famous Cities, increasingly boosts significant accumulation in them and puts unprecedented pressure on their protection. Though the accumulated functions and growing populations impede the conservation and renewal of these cities, decentralization of functions and population dispersion of the ancient towns are hard to realize, as scientific planning is often lacking when it comes to complicated situations. The HUL approach and the concept of sustainable tourism raised by UNESCO and WHC provide a new insight and urge us to jump out of the traditional idea of static protection, consider the sustainable tourism development of ancient towns and care about urban landscape protection and urban development [46].

Given the increasing attention concerning the balance between the conservation and development of historic cities, there is some lack of valid primary research for specific plans and those for practice. Therefore, it is a challenge faced by both urban planners and local governments to build a modern city while retaining its cultural diversity. This study would like to fill in a gap in the research and propose a primary research model for urban decentralization. Firstly, the review on past and current policies for urban decentralization in the West and Asia highlights the necessity of top-level planning and the orientation of the urban function, while the literature of HUL and sustainable planning emphasize the importance of participative approaches. Comparing the experiences between metropolis and small-scale cities, there are several common points [47,48].:

- Always relying on a certain theory or research;
- Respecting the concept of 'Planning Runs First';
- Focusing on land use and the adaptation of infrastructures;
- Regional understanding of the orientation of the development of ancient towns.

In this paper, we are also eager to apply public participation to the top-level development of the ancient towns and try to understand residents' opinions towards tourism. Taking Tingchow as a case study, many studies and surveys have been conducted before planning, underlining public participation and multi-body cooperation. The results showed that residents showed a positive attitude towards the impacts of tourism. Therefore, non-famous urban functions should be decentralized, and the findings of this study also indicate that residential population should account for one fifth to one third of the total population, and tourist population should be kept between 80,000 and 120,000. It will not only improve living conditions and promote economy, but also prevent the damage of historic relics and environment due to over-exploitation.

Admittedly, sustainable tourism development is not about creating an amusement-park-like environment, population control is not the ultimate goal, it just provides an alternative space for restoration and regeneration of the historic urban landscape. Furthermore, the methodology outlined in the study may be useful for other emerging historic tourism cities where the low life quality of residents brought by congestion is a serious problem and where tourism development and growing tourists are inevitable. Finally, though this research is not perfect, since the theory of dynamic renovation lacks a complete explanation, we still provide certain guidance for planners and relevant workers to work on the decentralization of ancient towns.

**Author Contributions:** Conceptualization, Y.M. and X.J.; methodology, Y.M.; investigation, Y.M. and Y.Z..; writing—original draft preparation, Y.M.; writing—review and editing, Y.M. and Y.Z.

**Funding:** This research is supported by Chinese National Natural Science Foundation (51278239,40871296). In addition, simultaneously, after these comprehensive studies, our group, the School of Architecture and Urban Planning of Nanjing University began to carry out the Master Plan of Tingchow (2016–2030), Planning on Conservation of Historic and Cultural Famous Cities in Tingchow County, Fujian Province (2016–2030) and Tourism Development Planning in Tingchow County, Fujian Province (2016–2030), which continues until now.

**Acknowledgments:** I am extremely grateful to Housing and Construction Bureau of Tingtow, Historic City Management Committee of Tingtow (NGO) and Planning and Research Studio of Smart City, Nanjing University, who provided data and gave me a great help in data processing.

**Conflicts of Interest:** The authors declare no conflict of interest.

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
