# Peer review of "Preliminary Research on Planning of Decentralizing Ancient Towns in Small-Scale Famous Historic and Cultural Cities with a Case Study of Tingchow County, Fujian Province"

_sustainability, doi:10.3390/su11102911_

Round 1
Reviewer 1 Report
Although I remain sceptical about the need to decentralize and ultimately displace residents from this and other historic towns and cities, especially given the underlying intention of promoting tourism growth (the antithesis of sustainable tourism), the authors I believe do provide a useful methodology that could potentially be applied to other congested historic cities where decentralization policy is in place. To begin, the authors provide sufficient background information on international as well as Chinese policy towards heritage conservation. They also offer an overview of how the destruction of historic landscapes remains a problem in China with uncontrolled growth, before introducing the concept of “Historic Urban landscapes.” The review on past and current policies for urban decentralization in the West and Asia provides additional context for the study as does the methodology section with reference to projected tourism growth and public awareness/acceptance of further tourism development based on a survey. The paper also provides an array of useful tables and figures that complement the text. The reference list is sufficient and it is clear that much work had been put into the research behind the study.
With regards to the writing, the authors need to further edit and proofread their manuscript. For example, note the sharp transition on page 2, line 50 (from an explanation of “famous cities” to human migration), awkward sentences where the meaning seems to be lost (e.g. p.4 line 172), typos (e.g. data, p. 6., line 233; Master Plan, p. 11, line 390; adopts, p. 17, line 486), and incorrect words (e.g. harmonic areas, p. 15, line 451). Finally, the authors should consider acknowledging that sustainable tourism development is not about creating an amusement park-like environment and that while decentralization may be necessary in Tingchow, the displacement of residents specifically for tourism development purposes is not the goal. Rather, the methodology outlined in the study may be useful for other emerging historic tourism cities where the quality of life of residents due to congestion is a serious problem and where tourism growth and growing tourist numbers is likely inevitable.
Author Response
Point 1: With regards to the writing, the authors need to further edit and proofread their manuscript. For example, note the sharp transition on page 2, line 50.
Response 1: I have asked professional agency for extensive editing of English language and corrected some words, and rewritten Paragraph 2. However, harmonic areas is a kind of proper nouns, which I haven't found a suitable alternative of it.
Point 2: conclusions can be improved
Response 2: I have added some literature review and revised the conclusion. Besides,I use some ideas from your comments. If you feel uncomfortable, please let me know.

Reviewer 2 Report
The aim of paper Is interesting and promising in terms of providing guidance on Urban decentralization. I find weaknesses in a critical letterature review (Lines 124-173) and a systematic conclusion.
Introduction Is too long, I found a lot material not being rlevant to the current argument.
Paragraph 2 appear not appropriate because It concern China but It would help to understand the argument more clearly and articolate the existing literature on urbanization and Urban functions.
Methodoogy and framework appear appropriate but would be more schematic and systematic.
The merit of paper Is diminished in part due to text and a short conclusion. In short, the authors rewrite the literature review, schematic and synthesis your results. There Was a weak link between paragraphs and please have a look at the systematic Analysis of data.
Author Response
Point 1: Introduction Is too long, I found a lot material not being rlevant to the current argument. Paragraph 2 appear not appropriate because It concern China but It would help to understand the argument more clearly and articolate the existing literature on urbanization and Urban functions.
Response 1: Some sentences in introduction part have been deleted and I have rewritten Para.2
Point 2: Methodoogy and framework appear appropriate but would be more schematic and systematic.The merit of paper Is diminished in part due to text and a short conclusion. There Was a weak link between paragraphs and please have a look at the systematic Analysis of data.
Response 2: I have rewritten conclusion and literature review though reading more references. Besides,I have asked professional agency for extensive editing of English language and corrected some awkward sentence between paragraphs.

Round 2
Reviewer 2 Report
The new version of manuscript is ok.